# A 30-Day Randomized Crossover Human Study on the Safety and Tolerability of a New Micellar Berberine Formulation with Improved Bioavailability

**DOI:** 10.3390/metabo15040240

**Published:** 2025-04-01

**Authors:** Afoke Ibi, Chuck Chang, Yun Chai Kuo, Yiming Zhang, Min Du, Yoon Seok Roh, Roland Gahler, Mary Hardy, Julia Solnier

**Affiliations:** 1ISURA, Clinical Research, Burnaby, BC V3N4S9, Canada; aibi@isura.ca (A.I.); cchang@isura.ca (C.C.); yzhang@isura.ca (Y.Z.); mdu@isura.ca (M.D.); kroh@isura.ca (Y.S.R.); 2Factors Group R & D, Burnaby, BC V3N4S9, Canada; 3Academy of Integrative and Holistic Medicine, San Diego, CA 92037, USA; mary@maryhardy.com

**Keywords:** berberine, bioavailability, delivery system, human study, LipoMicel, micelles, safety, tolerability, crossover study, blood chemistry, hepatoprotective

## Abstract

**Background/Objectives:** Berberine is a naturally occurring compound found in several plants and has been traditionally used for its various health benefits. However, its poor bioavailability limits its therapeutic potential. Berberine LipoMicel^®^ is a novel micellar formulation of berberine, microencapsulated within an emulsified matrix, designed to enhance bioavailability and bioactivity. This study aims to evaluate its safety, ensuring that improved bioavailability does not introduce new safety concerns. **Methods**: To assess its safety, a randomized, double-blind, placebo-controlled crossover study with a minimum 4-week washout period was conducted in 19 healthy participants over 30 days. The participants received 1000 mg of the treatment daily (i.e., 2 capsules/d), and their capillary blood was analyzed every week to monitor for changes in established safety markers related to liver and kidney function, including aspartate aminotransferase (AST), alanine aminotransferase (ALT), total bilirubin (TB), creatinine, fasting glucose (GLU), HbA1c, and various electrolytes. Additionally, potential side effects were recorded through the collection of weekly health questionnaires to determine treatment tolerability. **Results**: Compared to placebo, no statistically significant changes in any of the safety markers related to liver or kidney health were detected. Within-group analysis revealed a significant reduction of total cholesterol (TC) in females after 30 days of Berberine LipoMicel^®^ treatment. Although not significant, both male and female participants showed a noticeable improvement in the mean AST, potentially signaling a hepatoprotective effect. As for tolerability, no adverse events were reported by any of the participants. **Conclusions**: Based on these findings, despite higher bioavailability of berberine in a newly formulated delivery system (LipoMicel^®^), the treatment was found to be safe and well tolerated by human participants, with no significant deviations in blood chemistry that would indicate safety concerns over a period of 30 days.

## 1. Introduction

Plants have a long history of use in the form of powders, teas, poultices, infusions, and essential oils for their medicinal properties in various traditional cultures, and many modern-day natural health products consist of medicinal ingredients (MI) purified from traditional plants [1,2]. Berberine is one such medicinal ingredient, extracted from the root, stem or rhizome of various plants in the *Berberis*, *Coptis chinensis*, *Hydrastis*, and *Phellodendron* species [3]. For 3000 years, these berberine-containing herbs have been employed in traditional Ayurvedic and Chinese medicine to treat viral and bacterial infections, inflammation, and various ailments, such as dysentery, jaundice, and skin ulcers [4,5]. In more recent years, most of the research has focused on berberine’s effects on markers of glycemic control, blood lipids, and liver function in people with metabolic disorders [6,7,8]. Multiple studies have shown promising results in preventing metabolic and cardiovascular diseases in humans [9,10,11]. Regarding its effects on glycemic control, studies have shown that the glucose-lowering effects of berberine are similar to those of many drugs used to reduce blood sugar in individuals with Type 2 diabetes mellitus. This presents berberine as a possible, inexpensive and more accessible natural alternative to current glucose-reducing drugs [12].

While both short- and long-term uses of berberine are generally regarded as safe, with only a few side effects reported—primarily related to digestive issues, such as bloating, nausea, diarrhea, or constipation [13]—regulatory oversight varies widely across countries. For instance, in Canada, berberine is classified as a natural health product (NHP) and is permitted for sale as such without major restrictions. In the United States, it is marketed as a dietary supplement under the Dietary Supplement Health and Education Act (DSHEA) where manufacturers are responsible for ensuring safety. In contrast, European regulations are more fragmented. Given these variations, as well as the increasing development of novel formulations to enhance the bioavailability of berberine (such as LipoMicel^®^), more in-depth safety studies are necessary to fully evaluate the risks and tolerability of this compound in humans.

Berberine exhibits a broad spectrum of biological activities, influencing various physiological systems, particularly liver and kidney functions. It supports liver health by modulating enzyme activity and influences kidney function through its regulation of blood glucose and lipid metabolism, which may, in turn, affect electrolyte balance by altering excretion or retention [14,15,16]. However, when berberine is administered at high doses or potentially in highly bioavailable formulations, these effects may be amplified. While existing studies suggest that berberine can offer benefits for liver health, there is a concern that higher concentrations or prolonged use could lead to hepatic toxicity, particularly in individuals with pre-existing liver conditions [17]. Therefore, comprehensive safety studies, such as the present one, are crucial to assess the potential risks associated with higher doses and/or improved bioavailability, ensuring that the therapeutic benefits of berberine outweigh any possible harm.

There are many common and well-established blood markers that can be assessed to determine the safety profile of a drug-like or pharmacologically active compound like berberine. Measuring levels of proteins, such as bilirubin, and enzymes, such as aspartate aminotransferase (AST) and alanine aminotransferase (ALT), are good determinants of liver health [18]. Tests measuring levels of AST and ALT measure hepatocellular injury, while tests measuring total serum bilirubin (TB) levels aid in evaluating the state of the liver’s excretory function [19,20]. Notably, the elevation of serum levels of AST and ALT alone is not sufficient for verifying drug liability; some drugs have been reported to cause AST and ALT serum level elevations with no clinically significant hepatocellular injury [21]. As a result, recent FDA guidance has suggested the consideration of so-called Hy’s Law in clinical trials, where elevated levels of both serum ALT and serum bilirubin would indicate a potential drug-induced liver injury and a significant safety risk in participants—prompting further investigation [22].

Additional blood chemistry tests that can be performed to determine any safety concerns measure changes in levels of kidney function markers, such as serum creatinine and glomerular filtration rate (GFR), which are considered standard practice in clinical safety monitoring. Although GFR is not the most sensitive marker, it is a well-established marker and one of the best available clinical measures of kidney health. While GFR varies depending on factors such as age, weight, and gender, the average GFR in young adults ranges between 120 and 130 mL per minute per 1.73 m^2^. Although a gradual decline of approximately 1 mL per minute per year after the age of 30 is typical, a GFR less than 60 mL per minute per 1.73 m^2^ may indicate a loss of at least half of normal kidney function [23]. Given the relatively short 30-day observation period, monitoring for significant acute declines in GFR can be crucial for detecting compound-related kidney dysfunction. GFR is closely linked to changes in creatinine levels. Creatinine is a protein derived from creatine in skeletal muscles that is filtered and secreted through the proximal tubules of the kidneys. Creatinine clearance levels typically tend to exceed the GFR rate; thus, when GFR decreases, creatinine increases [23,24]. Accordingly, monitoring changes in creatinine levels alongside GFR can help provide a more accurate assessment of potential compound-related adverse effects.

Since previous studies have reported improved bioavailability, bioactivity (related to fasting blood glucose), and metabolite profile of berberine formulated in a new delivery system (LipoMicel^®^) without adverse events [25,26,27], the purpose of this research was to confirm that Berberine LipoMicel^®^ is safe and well tolerated for short-term use in humans and to further support the usage of berberine as a medicinal plant ingredient.

LipoMicel^®^ is a microemulsion-forming technology that solubilizes medicinal ingredients like berberine in oil matrices suitable for the manufacture of soft-gelatin capsules. The process of creating these LipoMicel^®^ emulsions makes diverse Natural Health Products (NHPs) easier to absorb compared to regular powder extracts. The hypothesis of this 30-day, placebo-controlled, double-blinded, crossover trial was that Berberine LipoMicel^®^ is safe and well tolerated for short-term use in humans, with no statistically significant or clinically meaningful deviations in blood safety markers when compared to the placebo group.

## 2. Materials and Methods

### 2.1. Ethics and Regulatory Approvals

The study protocol and amendments were reviewed by the Canadian Shield Ethics Review board with OHRP Registration IORG0003491, FDA Registration IRB00004157, under REB Tracking Number: 2022-11-002. The study was conducted in accordance with the ethical principles in the Declaration of Helsinki. Written informed consent was obtained from participants upon enrollment. The study has been registered on ClinicalTrials.gov with Identifier NCT06732908.

### 2.2. Objectives

The primary objective was to evaluate several safety parameters primarily related to liver and kidney function through blood biochemistry testing. Possible adverse events in healthy volunteers were recorded via the collection of health questionnaires over a 30-day period. Additionally, any potential health benefits or improvements were monitored over the 30-day period.

### 2.3. Treatments

Berberine LipoMicel^®^ containing 500 mg of berberine HCl as an MI and the following as non-medicinal ingredients (NMIs): medium-chain triglycerides, methylsulfonylmethane (MSM), stevia, and bergamot flavor. The softgel capsules were not modified from their commercial state. Participants were provided 60 soft-gelatin capsules, each containing 500 mg of berberine in a LipoMicel^®^ matrix. Two capsules were self-administered orally with water each day (i.e., 1000 mg/d) in the morning, along with breakfast. Softgel capsules remaining after the 30-day study from the container provided to each randomly selected participant were counted to monitor compliance. Placebo capsules containing microcrystalline cellulose were also administered to a second group of participants.

The formulation was tested according to standards set out in Health Canada’s Quality of Natural Health Products Guide, which includes Quality Control parameters, such as testing for lead, mercury, cadmium, arsenic, chromium, pesticide residues, solvent residues, microbial impurities, identity, and total berberine content.

### 2.4. Study Participants

#### 2.4.1. Inclusion Criteria

Participants were healthy men or women between 21 and 65 years of age. Upon study enrollment, participants were to complete an online health questionnaire on their medical history. A voluntarily signed informed consent form was a condition of enrollment.

#### 2.4.2. Exclusion Criteria

Participants who fell under the following criteria were excluded from the study:use of anti-inflammatory or non-steroidal anti-inflammatory medication;presence of cardiovascular disease and/or other acute or chronic diseases (e.g., liver, kidney or gastrointestinal diseases);use of cannabis, nicotine or tobacco;drinking an excess of alcohol (>20 g/day);those who are or plan to become pregnant;use of antioxidant supplements;use of cholesterol-lowering agents;participation in another investigational study.

### 2.5. Randomization and Blinding

The randomization sequence was generated using Microsoft Excel’s random number function and was securely stored to maintain allocation concealment until after data analysis. The study was conducted in two phases with a 1:1 allocation ratio of participants. Sequence randomization was applied by means of an online research randomization tool (https://www.randomizer.org, accessed on 24 May 2022). One member of the research team generated 19 sets of 1 s and 2 s, which were assigned to the list of participants provided. The remaining members of the research team, as well as each participant, were blinded to the code-to-treatment assignments in both Phases 1 and 2. Participants were assigned treatments in a randomized order, and both participants and study personnel were blinded to the assigned formulation. To ensure blinding, capsules were packaged in identical opaque bottles labeled with a study-specific code.

### 2.6. Study Design

This study used a randomized, double-blind, placebo-controlled crossover design. It was conducted from late spring to fall months (May to October 2024) in British Columbia, Canada. Participants were recruited from within the province and were then first assessed at Visit 1 (i.e., Study Week −1) to determine eligibility. There was a washout period of at least 4 weeks between interventions. Only healthy participants who met the eligibility criteria of the initial screening were included in the study. Then, baseline values were measured at Visit 2 (i.e., Study Week 0), and all subsequent measurements were compared against the baseline values to monitor any changes. Any change in blood chemistry parameters outside of normal variation was captured and assessed for potential health impacts due to the treatment. In Phase 1, randomly selected participants either took two softgel capsules of Berberine LipoMicel^®^ daily (1000 mg/day) or the same dose of placebo capsules (microcrystalline cellulose) for 30 days. Blood chemistry was monitored weekly, on day 1, day 7, day 14, day 21 and day 30. In Phase 2, after a sufficient washout period, the study was repeated for another 30 days, with all the participants who took the placebo in Phase 1 now being administered berberine capsules and vice versa (Figure 1). Safety parameters, such as liver (AST, ALT, TB) and kidney (creatinine, eGFR) markers, were monitored. Blood samples were also analyzed for fasting blood glucose (GLU), average blood glucose (by way of HbA1c) and blood lipids (high-density lipoprotein (HDL), low-density lipoprotein (LDL), total cholesterol (TC), and triglycerides (TG)) as well as the following electrolytes: calcium, phosphate, magnesium, potassium, sodium, and chloride. Total carbon dioxide (tCO_2_ or serum bicarbonate HCO_3_^−^) was also measured alongside electrolytes. Participants were instructed to arrive for assessment after an 8 h fast. After their fasting blood sample was taken for analysis, they were allowed to take the corresponding dose for the day. All blood samples were tested on the same instrument with the same test kits. All adverse events were recorded throughout the study period via health questionnaires.

### 2.7. Sample Size

The sample size was calculated using GPower 3.1.9.7. The input parameters were a two-tailed analysis with an effect size of 1, an alpha error of 0.05, and a power of 0.80 (beta error of 0.20). This yielded a sample size of 10 participants.

In addition, pre-trial investigations provided further calculations based on specific biomarkers: for HbA1c (normal value: 5.08, SD 0.46, abnormal value: 6.11, SD 0.46) with a correlation of 0.5, the effect size was 2.24, leading to a required sample size of 4; for alanine aminotransferase (ALT) (normal value: 27.6, SD 11.1, abnormal value: 41, SD 11.1) with a correlation of 0.5, the effect size was 1.21, yielding a sample size of 8; for glomerular filtration rate (GFR) (normal value: 120.4 SD 11.3, abnormal value: 89, SD 11.3), the effect size was 2.78, resulting in a sample size of 4.

### 2.8. Blood Chemistry Analysis

Capillary blood samples were analyzed using the SD-1 Auto Dry Biochemistry Analyzer (Seamaty Technology Co., Ltd., Chengdu, China) to assess several well-established safety markers, including liver and kidney function parameters, electrolytes, and other relevant indicators. On each test day, approximately 300 µL of capillary blood was collected from each participant using a standardized fingerstick method to ensure consistency across samples.

Immediately after collection, the samples were transferred into lithium–heparin-coated micro vials (Sarstedt AG & Co. KG, Nümbrecht, Germany, reference #: 20.1292.100) to prevent coagulation. For HbA1c analysis, a 10 µL aliquot of each blood sample was diluted with a lysis buffer (Seamaty Technology Co., Ltd., Chengdu, China) to facilitate hemoglobin release and ensure accurate measurement.

All blood samples were analyzed immediately after collection to maintain sample integrity and minimize pre-analytical variability. The Seamaty SD-1 Analyzer was calibrated according to the manufacturer’s standards to ensure the reliability of results.

### 2.9. Statistical Analysis

A statistical analysis of both male and female groups was performed. Intention-to-treat (ITT) analysis was performed on the presented data. Significance in changes from baseline response was determined with student’s *t*-tests for between-group comparisons and repeated measures ANOVA with multiple comparisons correction for within-group analysis across the study period.

### 2.10. Adverse Event (AE) Reporting

Participants were instructed to document all adverse events that occurred during the treatment periods via weekly collected online health questionnaires and report these (based on severity) immediately to the research team. Criteria for AE rating are shown in Appendix A.

## 3. Results

### 3.1. Demographic and Baseline Characteristics

The demographic and baseline characteristics of each participant’s blood chemistry (Table 1) were recorded at baseline, week 0, before treatment started to determine eligibility. 19 participants (12 healthy males and 7 females; average age ~35 years old) were included in a randomized, placebo-controlled crossover safety trial; all safety markers were within the range upon conclusion. Eighteen participants completed the entire study period (Figure 1).

### 3.2. Blood Chemistry

As displayed in Table 2 and Table 3, all tested markers stayed within recommended limits and no statistically significant or clinically meaningful deviations were observed. In both groups (male and female), certain markers like AST saw a gradual decline while remaining within the recommended range for a healthy patient. Although non-significant, there was a noticeable decrease in blood lipids, such as TG and TC, in the female participants, as well as in fasting blood glucose and HbAc1 over 30 days. Male participants experienced a slight reduction in LDL over 30 days. HbA1c data showed slight variations for female participants, which may be related to hormonal fluctuations, although this has not been controlled for in the current trial. The mean total carbon dioxide (tCO_2_) saw a statistically significant increase in the female group, although without any clinically meaningful deviation.

As demonstrated in Table 4 and Table 5, there were no significant differences between the berberine treatment group and the placebo group across both genders, except for fasting blood glucose and HbA1c, which were significantly lower in the female placebo group at 30 days. Most importantly, across both groups, all markers stayed within the healthy range. There was a significant decrease in total cholesterol (4.2 mmol/L to 3.8 mmol/L, *p* = 0.0036) in female participants following the berberine treatment after 30 days. Overall, no significant safety issues were detected in either gender group based on the tested blood markers.

### 3.3. Adverse Events

As shown in Table 6, no adverse events (AEs) were observed. Therefore, the berberine treatment was well tolerated in study participants.

## 4. Discussion

Based on previous investigations showing higher bioavailability of Berberine LipoMicel^®^ [26], the goal of this study was to evaluate its safety and tolerability for short-term use in humans, specifically assessing potential adverse events and changes in safety parameters over a 30-day period. No statistically significant or clinically meaningful deviations in any of the safety markers were observed over the 30-day study period. Furthermore, no adverse events were reported by any participants, indicating that the treatment product is well tolerated. These results align with previous studies on berberine, which suggest no safety concerns. For example, a meta-analysis on the safety of berberine by Guo et al. also showed no toxic effects on serum creatinine and no increased risk of serious adverse events, with dosages ranging from 0.6 g to 1.5 g [28]. Another more recent 2024 review by Nie et al. showed that berberine exhibited a favorable safety profile and significantly reduced ALT (mean difference (MD) = 0.72, *p* < 0.00001), AST (MD = −0.79, *p* < 0.0001), TG (MD = −0.59, *p* < 0.0001), LDL (MD = 0.53, *p* = 0.003), and TC (MD = −0.74, *p* < 0.00001) [29].

Interestingly, female participants in the placebo group had significantly lower fasting glucose and HbA1c concentrations compared to those receiving berberine treatment. A possible explanation for this could be a carryover effect, suggesting that the washout period may not have been sufficient to fully eliminate the effects of berberine before participants crossed over to the placebo phase. Given berberine’s potential to influence glucose metabolism over time, residual effects from the treatment phase could have contributed to the observed glucose and HbA1c differences.

In the berberine treatment group, female participants showed a statistically significant increase in tCO_2_ from week 1 to week 4 (20.9 mmol/L to 22.9 mmol/L, *p* = 0.0174), moving towards the normal range and suggesting a potential improvement in acid-base homeostasis; however, there was no statistically significant difference compared to placebo. Furthermore, when examining within-group changes in the male and female berberine treatment groups, no significant changes were observed in any of the tested markers from baseline across the treatment weeks. These findings, combined with the absence of clinically relevant changes, further support the overall safety and tolerability of Berberine LipoMicel^®^.

Although not significant, noticeable decreases in blood lipid levels were detected in both genders (male and female) over 30 days, suggesting that berberine may have beneficial lipid-lowering effects, consistent with its known mechanisms of action. For example, female participants had lower levels of TG and TC, while male participants experienced lower concentrations of LDL over 30 days of berberine treatment. The differences between male and female participants may reflect sex-specific variations in lipid metabolism, possibly influenced by hormonal factors. However, the 30-day duration may have been insufficient to observe significant changes, and individual metabolic variability could also play a role.

Notably, TC levels in the female group showed a significant reduction from baseline to 30 days. Multiple studies have already demonstrated the significant cholesterol-lowering effects of berberine (both unformulated berberine extract and berberine chloride)—highlighting it as a promising natural agent for managing dyslipidemia [30,31]. Among the suggested pharmacological mechanisms of action is the up-regulation of LDL receptors through mRNA stabilization, which is distinct from statins [32], as well as the inhibition of cholesterol absorption in the intestine, i.e., by interfering with micellization, decreasing uptake and permeability, and inhibiting acyl-coenzyme A cholesterol acyltransferase-2 expression [33]. While not observed in the current study, potentially due to the very small female sample size (*n* = 7), a few studies reported that berberine may exert greater female-specific effects in terms of increases in HDL cholesterol [34].

Regarding blood sugar modulation, it should be noted that HbA1c typically measures the 3-month averages of blood sugar levels, whereas the changes observed in the participants were over a week-to-week basis spanning only a month. Therefore, HbA1c cannot reliably reflect glycemic improvement in 30 days. Nevertheless, female participants showed a noticeable downward trend in HbA1c compared to placebo after 30 days. In contrast, male participants did not exhibit the same trend in blood sugar reduction. This aligns with a recent meta-analysis, which concluded that berberine’s effects on glycemic control and insulin sensitivity may be more pronounced in women compared to men [35]. This may be attributed to estrogen’s role in enhancing insulin sensitivity, differences in fat distribution, and gender-specific gut microbiota composition. Other potential explanations include the fact that dietary habits or lifestyle changes were not controlled for in this study. Additionally, it is important to note that all participants were healthy, normal-weight individuals, and therefore, reductions in HbA1c (as well as blood lipids) may not have been as pronounced as in previous studies on (standard) berberine, which primarily focused on overweight populations over a longer study period. Furthermore, while in a previous study comparing different berberine treatments (i.e., standard vs. LipoMicel^®^ formulation), immediate effects on fasting blood glucose (~12% reduction) were observed within just 2 days of LipoMicel^®^ treatment in healthy participants, this was likely due to the treatment’s enhanced bioavailability leading to improved insulin sensitivity and glucose uptake [25]. Similarly, in the present study, a substantial 15% decrease in fasting blood glucose (from 6.0 mmol/L to 5.1 mmol/L) was observed in female participants within the first week of treatment, indicating a meaningful early metabolic improvement. This reduction was sustained over 30 days and accompanied by an early trend toward improvement in HbA1c. However, the changes did not reach statistical significance, likely due to the small sample size. Notably, the magnitude and timing of the observed fasting glucose reduction are comparable to the early-phase glycemic response typically seen with metformin initiation [36,37]. In contrast, standard berberine formulations generally require a longer duration and higher doses to achieve similar effects, highlighting the potential advantage of the LipoMicel^®^ formulation in delivering faster glycemic benefits.

Notably, although not significant, both berberine treatment groups (male and female) showed a noticeable improvement in AST levels across the treatment weeks (week 0 to week 4; Table 2 and Table 3). This finding is supported by previous studies, where berberine administered to mice induced with carbon tetrachloride (CCl_4_) to trigger liver damage was shown to suppress the rise in ALT and AST levels in a concentration-dependent manner [38,39]. This improvement was more pronounced in male participants than in females when compared to the placebo, which may suggest a hepatoprotective effect of berberine. The sex-specific differences may be attributed to liver enzyme activity, metabolism, and hormonal influence. Men generally have higher baseline AST levels and are more susceptible to oxidative stress, which could make them more responsive to berberine’s hepatoprotective effects. Additionally, differences in berberine metabolism and gut microbiota composition between sexes may contribute to these observations.

While the placebo-controlled, blinded, crossover design is a key strength of this study—allowing each participant to serve as their own control and reducing inter-individual variability—it also has several limitations. The relatively small sample size (*n* = 19), with a male-dominant cohort, may have contributed to observed sex-specific effects, such as the more pronounced hepatoprotective response in males. Additionally, differences in capsule appearance due to the nature of the LipoMicel^®^ softgel formulation may have affected blinding integrity. Although participants were blinded to the specific formulation and treatment status, the placebo was administered in a hardgel capsule, while Berberine LipoMicel^®^ was in a softgel capsule, which could have influenced participant perception or tolerability assessments.

Moreover, the crossover design introduces the potential for carryover effects, particularly given the variability in berberine metabolism despite the implemented washout period. The study also relied on standard blood chemistry panels, which may not have captured subtle or long-term safety effects. Lastly, the relatively healthy study population may limit generalizability to individuals with pre-existing conditions or those using berberine for therapeutic purposes. Despite these limitations, this study provides important safety data on Berberine LipoMicel^®^, supporting its potential as a novel, bioavailable formulation.

Future studies of Berberine LipoMicel^®^ would benefit from a potentially larger sample size and a more diverse demographic to better capture these potential variations. Additionally, extending the investigation period beyond 30 days could provide further insights into the long-term safety and efficacy of the treatment. Given that berberine is primarily indicated for the prevention or treatment of metabolic disorders such as type 2 diabetes, which are more prevalent in overweight or obese populations, future safety evaluations in these individuals would be valuable.

Beyond its potential safety and efficacy advantages, Berberine LipoMicel^®^ enhanced bioavailability may offer economic benefits by allowing for lower dosing while achieving similar or superior therapeutic effects compared to standard berberine formulations. Traditional berberine supplements often require high doses (e.g., 1000–1500 mg per day) due to poor absorption, which can increase cost and dosing burden for consumers. In contrast, improved absorption with Berberine LipoMicel^®^ may reduce the required dosage, potentially enhancing cost-effectiveness and patient compliance.

## 5. Conclusions

Overall, Berberine LipoMicel^®^ was well tolerated over the 30-day study period, with no reported adverse events. All tested markers remained within recommended limits, and no statistically significant or clinically meaningful deviations were observed in blood chemistry. These findings provide strong evidence that the enhanced bioavailability of the LipoMicel^®^ delivery system does not compromise safety in healthy individuals.

Additionally, preliminary sex-specific differences in metabolic responses were observed. Female participants had significantly lower total cholesterol (TC) levels after 30 days of Berberine LipoMicel^®^ treatment. Although not significant, male participants exhibited a greater reduction in LDL and AST levels, while female participants showed more pronounced improvements in triglycerides, fasting blood glucose and HbA1c. These trends align with existing research suggesting that berberine’s metabolic effects may vary between genders due to hormonal and physiological differences.

These findings reinforce the safety profile of Berberine LipoMicel^®^ and suggest potential metabolic benefits that warrant further investigation. While this study provides important initial data, further long-term, large-scale studies are needed to confirm these sex-specific responses and explore their clinical implications.

## Figures and Tables

**Figure 1 metabolites-15-00240-f001:**
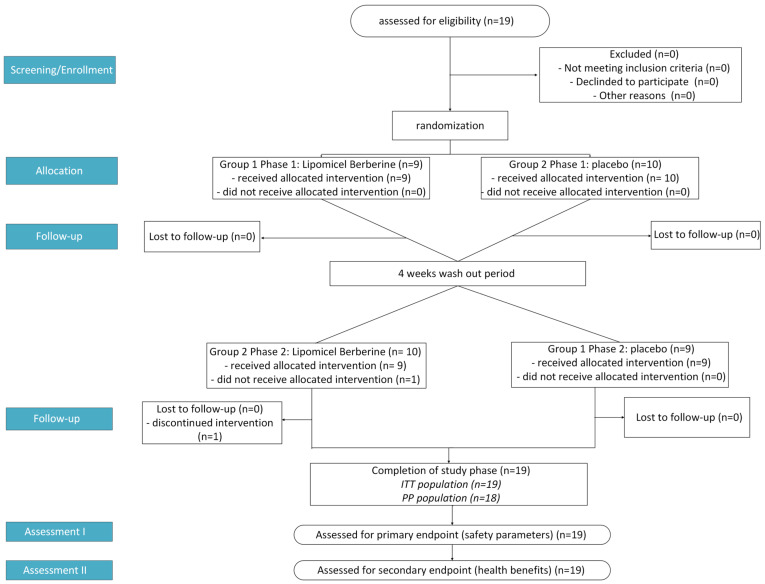
Flowchart showing study design process where *n* = number of participants, ITT = Intention to Treat, and PP = Per Protocol.

**Table 1 metabolites-15-00240-t001:** Demographic and baseline characteristics of participants.

		Males	Females	Combined
Number of participants		12	7	19
Age (years)		38.2 ± 8.9	31.6 ± 8.9	34.9 ± 9.1
Weight (kg)		74.5 ± 10.5	63.3 ± 12.8	68.9 ± 12.3
Height (cm)		173.8 ± 5.7	162.6 ± 7.7	168.2 ± 8.3
BMI		24.7 ± 3.2	24.2 ± 2.8	24.5 ± 3.1
Smokers		0	0	0
Medication		0	0	0
	Normal range			
TB (μmol/L)	3.4–21.0	18.3 ± 7.3	16.0 ± 8.5	17.2 ± 7.9
AST (U/L)	13–40	37.7 ± 20.2	28.7 ± 12.7	33.2 ± 16.5
ALT (U/L)	7.0–50	33.2 ± 18.8	20.3 ± 8.5	26.8 ± 13.6
Creatinine (μmol/L)	35.0–97.0	67.6 ± 18.0	65.2 ± 6.0	66.4 ± 12.0
eGFR (mL/min/1.73 m^2^)	>90	119.2 ± 11.0	111.3 ± 13.4	109.0 ± 11.2
HDL (mmol/L)	1.16–1.42	1.3 ± 0.2	1.6 ± 0.3	1.5 ± 0.2
LDL (mmol/L)	0.50–3.14	3.0 ± 0.7	2.0 ± 0.1	2.5 ± 0.4
TC (mmol/L)	0–5.17	5.0 ± 0.8	4.2 ± 0.3	4.6 ± 0.5
TG (mmol/L)	0–1.70	1.4 ± 0.6	1.2 ± 0.3	1.3 ± 0.4
HbA1c (mmol/L)	3.89–6.11	5.9 ± 0.8	6.7 ± 0.8	6.3 ± 0.8
GLU (mmol/L)	3.89–6.11	5.7 ± 0.5	6.0 ± 1.3	5.8 ± 0.9
tCO_2_ (mmol/L)	22.0–29.0	21.9 ± 1.5	20.2 ± 2.1	21.1 ± 1.8
Ca (mmol/L)	2.0–2.58	2.6 ± 0.1	2.5 ± 0.2	2.5 ± 0.1
P (mmol/L)	0.85–1.51	1.1 ± 0.2	1.1 ± 0.1	1.1 ± 0.1
Mg (mmol/L)	0.65–1.25	0.9 ± 0.1	0.8 ± 0.1	0.9 ± 0.1
K (mmol/L)	3.40–5.30	4.7 ± 0.4	4.7 ± 0.3	4.7 ± 0.3
Na (mmol/L)	135–147	139.7 ± 1.7	138.9 ± 3.3	139.3 ± 2.5
Cl (mmol/L)	99.0–112.0	105.8 ± 1.6	104.7 ± 5.0	105.3 ± 3.3

**Table 2 metabolites-15-00240-t002:** Statistical analysis of safety markers in males following the berberine treatment ^1^.

Male	Mean ± SD					*p*-Value	Normal Range
	Week 0	Week 1	Week 2	Week 3	30 Days		
Total Bilirubin (µmol/L)	18.3 ± 7.3	20.4 ± 7.4	15.7 ± 5.9	16.3 ± 5.2	17.1 ± 5.5	0.247	3.4–21.0
AST (U/L)	37.7 ± 20.2	32.2 ± 19.5	29.6 ± 12.7	23.6 ± 5.0	26.5 ± 7.4	0.178	15.0–40.0
ALT (U/L)	33.2 ± 18.8	31.2 ± 15.1	30.2 ± 15.4	30.9 ± 14.3	31.8 ± 12.1	0.498	9.0–50.0
Creatinine (µmol/L)	67.6 ± 18.0	75.1 ± 14.9	70.7 ± 14.4	72.4 ± 9.7	74.2 ± 13.4	0.362	44.0–97.0
eGFR (mL/min/1.73 m^2^)	119.2 ± 11.0	112.8 ± 15.4	115.3 ± 12.5	114.1 ± 13.1	112.2 ± 14.6	0.467	>90
HDL (mmol/L)	1.3 ± 0.2	1.3 ± 0.2	1.3 ± 0.3	1.3 ± 0.3	1.3 ± 0.3	0.689	1.16–1.42
LDL (mmol/L)	3.0 ± 0.7	2.8 ± 0.6	2.5 ± 0.8	2.6 ± 0.5	2.8 ± 0.6	0.123	0.50–3.14
TC (mmol/L)	5.0 ± 0.8	4.8 ± 0.8	4.5 ± 0.8	4.6 ± 0.6	4.7 ± 0.8	0.091	0–5.17
TG (mmol/L)	1.4 ± 0.6	1.4 ± 0.8	1.4 ± 0.8	1.6 ± 0.9	1.4 ± 0.7	0.674	0–1.70
HbA1c (mmol/L)	5.9 ± 0.8	6.4 ± 1.8	6.2 ± 1.2	6.0 ± 1.2	6.4 ± 0.9	0.437	3.89–6.11
GLU (mmol/L)	5.7 ± 0.5	5.5 ± 0.9	5.2 ± 0.5	5.5 ± 0.6	5.5 ± 0.6	0.197	3.89–6.11
tCO_2_ (mmol/L)	21.9 ± 1.5	22.7 ± 1.5	23.1 ± 1.0	23.4 ± 1.4	21.8 ± 3.6	0.295	22.0–29.0
Ca (mmol/L)	2.6 ± 0.1	2.6 ± 0.1	2.6 ± 0.1	2.6 ± 0.1	2.4 ± 0.7	0.355	2.00–2.58
P (mmol/L)	1.1 ± 0.2	1.1 ± 0.1	1.1 ± 0.1	1.2 ± 0.2	1.1 ± 0.3	0.482	0.85–1.51
Mg (mmol/L)	0.9 ± 0.1	0.9 ± 0.1	0.8 ± 0.0	0.9 ± 0.1	0.8 ± 0.0	0.411	0.65–1.25
K (mmol/L)	4.7 ± 0.4	4.8 ± 0.6	4.6 ± 0.3	4.7 ± 0.2	4.6 ± 0.3	0.235	3.40–5.30
Na (mmol/L)	139.7 ± 1.7	138.9 ± 1.4	139.6 ± 1.6	138.3 ± 1.4	133.9 ± 16.9	0.365	135.0–147.0
Cl (mmol/L)	105.8 ± 1.6	105.6 ± 1.6	105.5 ± 1.3	104.8 ± 1.2	105.3 ± 1.6	0.331	99.0–122.0

^1^ Data are expressed as the mean ± standard deviation. One-way ANOVA with a post hoc test (Bonferroni Correction) was performed, and the *p*-values represent statistical significance across the 5 groups, where *p* < 0.05 indicates significant differences.

**Table 3 metabolites-15-00240-t003:** Statistical analysis of safety markers in females following the berberine treatment ^1^.

Female	Mean ± SD					*p*-Value	Normal Range
	Week 0	Week 1	Week 2	Week 3	30 Days		
Total Bilirubin (µmol/L)	16.0 ± 8.5	13.3 ± 6.4	13.0 ± 4.8	13.1 ± 7.3	16.3 ± 9.8	0.783	3.4–21.0
AST (U/L)	28.7 ± 12.7	16.7 ± 2.8	18.6 ± 5.0	20.3 ± 9.0	20.6 ± 8.5	0.126	13.0–35.0
ALT (U/L)	20.3 ± 8.5	20.0 ± 10.8	19.9 ± 5.7	23.3 ± 15.1	26.1 ± 13.0	0.262	7.0–40.0
Creatinine (µmol/L)	65.2 ± 6.0	65.7 ± 5.7	64.3 ± 7.1	62.1 ± 8.8	58.7 ± 5.8	0.205	35.0–80.0
eGFR (mL/min/1.73 m^2^)	111.3 ± 13.4	110.4 ± 12.4	110.6 ± 12.3	113.3 ± 16.3	117.9 ± 7.4	0.516	>90
HDL (mmol/L)	1.6 ± 0.3	1.5 ± 0.3	1.6 ± 0.3	1.5 ± 0.3	1.4 ± 0.3	0.171	1.29–1.55
LDL (mmol/L)	2.0 ± 0.1	1.9 ± 0.3	1.9 ± 0.5	1.8 ± 0.3	2.0 ± 0.4	0.630	0.50–3.14
TC (mmol/L)	4.2 ± 0.3	4.0 ± 0.3	4.0 ± 0.4	3.8 ± 0.3	3.8 ± 0.4	0.162	0–5.17
TG (mmol/L)	1.2 ± 0.3	1.3 ± 0.6	1.2 ± 0.6	1.1 ± 0.2	1.0 ± 0.3	0.407	0–1.70
HbA1c (mmol/L)	6.7 ± 0.8	5.7 ± 1.0	6.2 ± 1.1	6.5 ± 1.4	6.3 ± 1.0	0.361	3.89–6.11
GLU (mmol/L)	6.0 ± 1.3	5.1 ± 0.6	5.0 ± 0.3	5.2 ± 0.6	5.1 ± 0.4	0.119	3.89–6.11
tCO_2_ (mmol/L)	20.2 ± 2.1	21.8 ± 1.1	22.5 ± 0.8	22.2 ± 0.8	22.9 ± 0.8	0.0174	22.0–29.0
Ca (mmol/L)	2.5 ± 0.2	2.4 ± 0.1	2.5 ± 0.1	2.5 ± 0.1	2.5 ± 0.1	0.595	2.00–2.58
P (mmol/L)	1.1 ± 0.1	1.2 ± 0.1	1.1 ± 0.2	1.2 ± 0.1	1.2 ± 0.1	0.108	0.85–1.51
Mg (mmol/L)	0.8 ± 0.1	0.9 ± 0.1	0.8 ± 0.1	0.8 ± 0.1	0.8 ± 0.1	0.591	0.65–1.25
K (mmol/L)	4.7 ± 0.3	4.9 ± 0.6	4.7 ± 0.3	4.8 ± 0.3	4.7 ± 0.4	0.560	3.40–5.30
Na (mmol/L)	138.9 ± 3.3	138.7 ± 2.6	139.2 ± 1.1	139.2 ± 1.6	139.3 ± 2.8	0.890	135.0–147.0
Cl (mmol/L)	104.7 ± 5.0	105.0 ± 4.0	107.1 ± 0.8	106.2 ± 2.0	106.2 ± 1.4	0.372	99.0–112.0

^1^ Data are expressed as the mean ± standard deviation. One-way ANOVA with a post hoc test (Bonferroni Correction) was performed, and the *p*-values represent statistical significance across the 5 groups, where *p* < 0.05 indicates significant differences.

**Table 4 metabolites-15-00240-t004:** Summary of treatment versus placebo male data ^1^**.**

Males	Berberine Group		Placebo Group		*p*-Value
	Baseline	30 days	Baseline	30 days	(Placebo vs. Berberine 30 days)
Total Bilirubin (µmol/L)	18.3 ± 7.3	17.1 ± 5.5	16.1 ± 5.2	16.8 ± 5.0	0.898
	*p* = 0.971		*p* = 0.73		
AST (U/L)	37.7 ± 20.2	26.5 ± 7.4	26.1 ± 7.5	24.8 ± 7.9	0.582
	*p* = 0.251		*p* = 0.676		
ALT (U/L)	33.2 ± 18.8	31.8 ± 12.1	33.7 ± 14.0	33.1 ± 10.0	0.787
	*p* = 0.977		*p* = 0.907		
Creatinine (µmol/L)	67.6 ± 18.0	74.2 ± 13.4	78.3 ± 13.0	70.9 ± 4.1	0.434
	*p* = 0.367		*p* = 0.072		
eGFR (mL/min/1.73 m^2^)	119.2 ± 11.0	112.2 ± 14.6	109.1 ± 15.3	116.2 ± 7.0	0.401
	*p* = 0.297		*p* = 0.156		
HDL (mmol/L)	1.3 ± 0.2	1.3 ± 0.3	1.4 ± 0.2	1.4 ± 0.2	0.507
	*p* > 0.9999		*p* = 0.911		
LDL (mmol/L)	3.0 ± 0.7	2.8 ± 0.6	2.8 ± 0.6	2.9 ± 0.7	0.765
	*p* = 0.752		*p* = 0.831		
TC (mmol/L)	5.0 ± 0.8	4.7 ± 0.8	5.0 ± 0.7	5.0 ± 0.8	0.508
	*p* = 0.794		*p* = 0.828		
TG (mmol/L)	1.4 ± 0.6	1.4 ± 0.7	1.8 ± 0.8	1.6 ± 0.9	0.557
	*p* = 0.715		*p* = 0.451		
HbA1c (mmol/L)	5.9 ± 0.8	6.4 ± 0.9	7.1 ± 3.2	6.3 ± 1.3	0.786
	*p* = 0.306		*p* = 0.378		
GLU (mmol/L)	5.7 ± 0.5	5.5 ± 0.6	5.5 ± 0.8	5.4 ± 0.7	0.562
	*p* = 0.403		*p* = 0.903		
tCO_2_ (mmol/L)	21.9 ± 1.5	21.8 ± 3.6	21.7 ± 1.6	23.0 ± 1.8	0.323
	*p* > 0.999		*p* = 0.060		
Ca (mmol/L)	2.6 ± 0.1	2.4 ± 0.7	2.6 ± 0.1	2.5 ± 0.1	0.459
	*p* = 0.809		*p* = 0.54		
P (mmol/L)	1.1 ± 0.2	1.1 ± 0.3	1.2 ± 0.2	1.2 ± 0.2	0.505
	*p* = 0.993		*p* = 0.930		
Mg (mmol/L)	0.9 ± 0.1	0.8 ± 0.0	0.8 ± 0.1	0.8 ± 0.1	0.382
	*p* = 0.350		*p* = 0.859		
K (mmol/L)	4.7 ± 0.4	4.6 ± 0.3	4.6 ± 0.3	4.7 ± 0.6	0.337
	*p* = 0.586		*p* = 0.529		
Na (mmol/L)	139.7 ± 1.7	133.9 ± 16.9	139.5 ± 1.1	137.8 ± 2.1	0.497
	*p* = 0.727		*p* = 0.031		
Cl (mmol/L)	105.8 ± 1.6	105.3 ± 1.6	106.3 ± 1.4	105.0 ± 2.0	0.346
	*p* = 0.678		*p* = 0.072		

^1^ Data are expressed as the mean ± standard deviation. *p*-values were generated from student’s *t*-tests, where *p* < 0.05 indicates significant differences.

**Table 5 metabolites-15-00240-t005:** Summary of treatment versus placebo female data ^1^.

Female	Berberine Group		Placebo Group		*p*-Value
	Baseline	30 days	Baseline	30 days	(Placebo vs. Berberine 30 days)
Total Bilirubin (µmol/L)	16.0 ± 8.5	16.3 ± 9.8	13.1 ± 5.8	17.7 ± 10.6	0.797
	*p* > 0.999		*p* = 0.340		
AST (U/L)	28.7 ± 12.7	20.6 ± 8.5	36.0 ± 46.0	17.6 ± 5.2	0.439
	*p* = 0.540		*p* = 0.719		
ALT (U/L)	20.3 ± 8.5	26.1 ± 13.0	19.5 ± 7.2	17.0 ± 4.8	0.105
	*p* = 0.383		*p* = 0.400		
Creatinine (µmol/L)	65.2 ± 6.0	58.7 ± 5.8	67.0 ± 8.2	64.7 ± 4.7	0.068
	*p* = 0.727		*p* = 0.930		
eGFR (mL/min/1.73 m^2^)	111.5 ± 17.0	117.9 ± 7.4	106.8 ± 10.8	120.9 ± 22.0	0.737
	*p* = 0.695		*p* = 0.469		
HDL (mmol/L)	1.6 ± 0.3	1.4 ± 0.3	1.6 ± 0.3	1.6 ± 0.3	0.161
	*p* = 0.0605		*p* > 0.999		
LDL (mmol/L)	2.0 ± 0.1	2.0 ± 0.4	2.0 ± 0.4	2.0 ± 0.6	0.888
	*p* = 0.974		*p* > 0.999		
TC (mmol/L)	4.2 ± 0.3	3.8 ± 0.4	4.3 ± 0.6	4.2 ± 0.6	0.136
	*p* = 0.0036		*p* = 0.995		
TG (mmol/L)	1.2 ± 0.3	1.0 ± 0.3	1.4 ± 0.8	1.3 ± 0.5	0.129
	*p* = 0.0728		*p* = 0.965		
HbA1c (mmol/L)	6.7 ± 0.8	6.3 ± 1.0	5.3 ± 0.6	5.1 ± 0.7	0.023
	*p* = 0.951		*p* = 0.917		
GLU (mmol/L)	6.0 ± 1.3	5.1 ± 0.4	5.0 ± 0.4	4.8 ± 0.3	0.0409
	*p* = 0.367		*p* = 0.724		
tCO_2_ (mmol/L)	20.2 ± 2.1	22.9 ± 0.8	22.2 ± 1.5	22.8 ± 0.5	0.749
	*p* = 0.0639		*p* = 0.551		
Ca (mmol/L)	2.5 ± 0.2	2.5 ± 0.1	2.5 ± 0.1	2.5 ± 0.1	> 0.999
	*p* = 0.998		*p* = 0.170		
P (mmol/L)	1.1 ± 0.1	1.2 ± 0.1	1.2 ± 0.1	1.1 ± 0.2	0.415
	*p* = 0.288		*p* = 0.709		
Mg (mmol/L)	0.8 ± 0.1	0.8 ± 0.1	0.8 ± 0.0	0.9 ± 0.1	0.369
	*p* = 0.997		*p* = 0.610		
K (mmol/L)	4.7 ± 0.3	4.7 ± 0.4	4.8 ± 0.3	4.6 ± 0.6	0.749
	*p* > 0.999		*p* = 0.779		
Na (mmol/L)	138.9 ± 3.3	139.3 ± 2.8	138.4 ± 2.1	139.8 ± 2.4	0.695
	*p* = 0.990		*p* = 0.314		
Cl (mmol/L)	104.7 ± 5.0	106.2 ± 1.4	105.4 ± 2.6	106.0 ± 2.6	0.829
	*p* = 0.773		*p* = 0.982		

^1^ Data are expressed as the mean ± standard deviation. p-values were generated from student’s *t*-tests, where *p* < 0.05 indicates significant differences.

**Table 6 metabolites-15-00240-t006:** Adverse events reported.

Safety Study Name:	Berberine LipoMicel^®^		
		*n*	percentage
Total number of participants (n)		18	
Participants reporting AE (n)		0	0%
Participants reporting type of AE (n)	Symptoms		
	Bloating	0	0%
	Constipation	0	0%
	Diarrhea	0	0%
	Heartburn	0	0%
	Pain, cramps, or a knotted feeling in your abdomen	0	0%
	Rash	0	0%
	Nausea	0	0%
	Dizziness	0	0%
	Blurred vision	0	0%
	Other (unrelated to treatment)	0	0%
Total AE by severity (n)			
	None	18	100%
	Mild	0	0%
	Moderate	0	0%
	Severe	0	0%
	Life-threatening	0	0%

## Data Availability

The original contributions presented in this study are included in the article. Further inquiries can be directed to the corresponding authors.

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
