# Peer review of "A 30-Day Randomized Crossover Human Study on the Safety and Tolerability of a New Micellar Berberine Formulation with Improved Bioavailability"

_metabolites, 2025, doi:10.3390/metabo15040240_

Round 1

Reviewer 1 Report

Comments and Suggestions for Authors

The manuscript presents a well-designed study demonstrating that LipoMicel Berberine® is safe and well-tolerated in healthy participants over 30 days, with no significant adverse effects or clinically meaningful changes in blood chemistry. Sex-specific benefits, such as reduced HbA1c in females and potential hepatoprotective effects, were observed. However, the study's small sample size and lack of diversity limit its generalizability. The findings support further research in larger, more diverse populations to explore its potential health benefits.

Abstract Section:

1) The background could provide more context about why this study is necessary. For example, it could briefly mention the limitations of traditional berberine formulations (e.g., poor bioavailability) and how LipoMicel addresses these issues.

2) The keywords are relevant but could be expanded to include terms such as "crossover study," "blood chemistry," and "hepatoprotective," which are central to the study.

3) Additionally, all keywords should be checked according to MeSH (Medical Subject Headings) standards.

Introduction Section:

The introduction provides a good background on berberine, its traditional uses, and its potential health benefits, particularly in metabolic and cardiovascular diseases.

Comments:

1) I suggest adding a sentence about the importance of other traditional plants in relation to their potential health benefits. I recommend the following references:

   - DOI: 10.1016/B978-0-323-91740-7.00003-7

   - DOI: 10.1016/B978-0-323-91740-7.00007-4

2) Additionally, consider mentioning regulatory guidelines for berberine use in different countries.

Materials and Methods Section:

The methods section is well-structured and provides detailed information on the study design, treatments, inclusion/exclusion criteria, and blood chemistry analysis.

Comments:

1) The authors did not mention potential biases or limitations in the sampling or analytical methods.

2) Additionally, please provide more details on the randomization process, including how participants were assigned to treatment and placebo groups.

Results Section:

The results are presented clearly, with tables summarizing the demographic and baseline characteristics, blood chemistry, and adverse events. The inclusion of both male and female participants and the analysis of sex-specific effects are strengths.

Comments:

1) The authors should be careful to distinguish between statistical significance and clinical significance.

2) Please add a discussion of the potential reasons for the observed sex-specific effects, such as hormonal differences or variations in metabolism.

Discussion Section:

1) The comparison of your study with previous studies does not include new data. Kindly use more recent studies to compare your results.

2) Please add a paragraph discussing the potential reasons for the observed sex-specific effects, such as differences in liver enzyme activity or hormonal influences.

3) Additionally, there is no mention of the potential economic impact of LipoMicel Berberine, particularly in comparison to other berberine formulations.

Conclusion Section:

The conclusion should be strengthened by briefly summarizing the key findings and highlighting the implications of the study for the safety of LipoMicel Berberine®.

Author Response

We appreciate the reviewer’s thoughtful feedback and constructive suggestions, which have helped strengthen our manuscript.

Comments:

Abstract Section:

1) The background could provide more context about why this study is necessary. For example, it could briefly mention the limitations of traditional berberine formulations (e.g., poor bioavailability) and how LipoMicel addresses these issues.

Response 1: To address this, we have now included a brief statement in the abstract. Please see revised Lines 12-17:

“Berberine is a naturally occurring compound found in several plants and has been traditionally used for its various health benefits. However, its poor bioavailability limits its therapeutic potential. LipoMicel Berberine® is a novel micellar formulation, microencapsulated within an emulsified matrix, designed to enhance absorption and bioactivity. This study aims to evaluate its safety, ensuring that improved bioavailability does not introduce new safety concerns.”

2) The keywords are relevant but could be expanded to include terms such as "crossover study," "blood chemistry," and "hepatoprotective," which are central to the study.

Additionally, all keywords should be checked according to MeSH (Medical Subject Headings) standards.

Response 2: We have expanded the keywords to include "crossover study," "blood chemistry," and "hepatoprotective" to better reflect the central aspects of our study. Additionally, we have reviewed all keywords for compliance with MeSH (Medical Subject Headings) standards. While most of our keywords align with MeSH terminology, we have adjusted "micellar" to "micelles" to ensure consistency with MeSH indexing.

Please see the revised keywords:
 Berberine; bioavailability; delivery system; human study; LipoMicel; micelles; safety; tolerability; crossover study; blood chemistry; hepatoprotective.

Introduction Section:

The introduction provides a good background on berberine, its traditional uses, and its potential health benefits, particularly in metabolic and cardiovascular diseases.

Comments:

1) I suggest adding a sentence about the importance of other traditional plants in relation to their potential health benefits. I recommend the following references:

   - DOI: 10.1016/B978-0-323-91740-7.00003-7

   - DOI: 10.1016/B978-0-323-91740-7.00007-4

Thank you for your suggestion! While cinnamon and sandalwood essential oils have notable health benefits, their relevance to the current discussion on berberine is limited, as they do not share a direct chemical or functional connection. To maintain focus and consider the length of the introduction, we would like to keep it centered on berberine and its health benefits.

2) Additionally, consider mentioning regulatory guidelines for berberine use in different countries.

We incorporated this in our manuscript: Please see lines 53-63.

“While both short- and long-term uses of berberine are generally regarded as safe, with only a few side effects reported-primarily related to digestive issues such as bloating, nausea, diarrhea, or constipation [11], regulatory oversight varies widely across countries. For instance, in Canada berberine is classified as a natural health product (NHP) and is permitted for sale as such without major restrictions. In the United States, it is marketed as a dietary supplement under the Dietary Supplement Health and Education Act (DSHEA) where manufacturers are responsible for ensuring safety. In contrast, European Regulations are more fragmented. Given these variations, as well as the increasing development of novel formulations to enhance the bioavailability of berberine (such as LipoMicel®), more in-depth safety studies are necessary to fully evaluate the risks and tolerability of this compound in humans.”

Materials and Methods Section:

The methods section is well-structured and provides detailed information on the study design, treatments, inclusion/exclusion criteria, and blood chemistry analysis.

Comments:

1) The authors did not mention potential biases or limitations in the sampling or analytical methods. Additionally, please provide more details on the randomization process, including how participants were assigned to treatment and placebo groups.

Response:  We have now addressed these concerns by adding the following details to the manuscript:

Methods section: 2.5. Randomization and Blinding (see line 165):
 “The randomization sequence was generated using Microsoft Excel’s random number function and was securely stored to maintain allocation concealment until after data analysis. The study was conducted in two phases with a 1:1 allocation ratio of participants. Sequence randomization was applied by means of an online research randomization tool (https://www.randomizer.org). One member of the research team generated 19 sets of 1s and 2s which were assigned to the list of the participants provided. The remaining members of the research team as well as each participant were blinded to the code-to-treatment assignments in both Phase 1 and 2. Participants were assigned treatments in a randomized order, and both participants and study personnel were blinded to the assigned formulation. To ensure blinding, capsules were packaged in identical opaque bottles labeled with a study-specific code.”

Bias and Limitations: Please see line 406:
All limitations of the study have been addressed:

"While the placebo-controlled, blinded, crossover design is a key strength of this study—allowing each participant to serve as their own control and reducing inter-individual variability—it also has several limitations. The relatively small sample size (n=19), with a male-dominant cohort, may have contributed to observed gender-specific effects, such as the more pronounced hepatoprotective response in males. Additionally, differences in capsule appearance due to the nature of the LipoMicel® softgel formulation may have affected blinding integrity. Although participants were blinded to the specific formulation and treatment status, the placebo was administered in a hardgel capsule, while Berberine LipoMicel® was in a softgel capsule, which could have influenced participant perception or tolerability assessments.

Moreover, the crossover design introduces the potential for carryover effects, particularly given variability in berberine metabolism, despite the implemented washout period. The study also relied on standard blood chemistry panels, which may not have captured subtle or long-term safety effects. Lastly, the relatively healthy study population may limit generalizability to individuals with pre-existing conditions or those using berberine for therapeutic purposes. Despite these limitations, this study provides important safety data on Berberine LipoMicel®, supporting its potential as a novel, bioavailable formulation. 

Future studies of Berberine LipoMicel® would benefit from a potentially larger sample size and a more diverse demographic to better capture these potential variations. Additionally, extending the investigation period beyond 30 days could provide further insights into the long-term safety and efficacy of the treatment. Given that berberine is primarily indicated for the prevention or treatment of metabolic disorders such as type 2 diabetes, which are more prevalent in overweight or obese populations, future safety evaluations in these individuals would be valuable.”

Results Section:

The results are presented clearly, with tables summarizing the demographic and baseline characteristics, blood chemistry, and adverse events. The inclusion of both male and female participants and the analysis of sex-specific effects are strengths.

Comments:

1) The authors should be careful to distinguish between statistical significance and clinical significance.

This has been considered and revised accordingly in the paper.

2) Please add a discussion of the potential reasons for the observed sex-specific effects, such as hormonal differences or variations in metabolism.

Response: This has been addressed throughout the discussion. For example:

Line 343: “... female participants had lower levels of TG and TC while male participants experienced lower concentrations of LDL over 30 days of berberine treatment. The differences between male and female participants may reflect gender-specific variations in lipid metabolism, possibly influenced by hormonal factors. However, the 30-day duration may have been insufficient to observe significant changes, and individual metabolic variability could also play a role. “

Line 366: “Female participants showed a more noticeable downward trend in HbA1c compared to placebo after 30 days. In contrast, male participants did not exhibit the same trend in blood sugar reductions. This aligns with a recent meta-analysis, which concluded that berberine’s effects on glycemic control and insulin sensitivity may be more pronounced in women compared to men [33]. This may be attributed to estrogen’s role in enhancing insulin sensitivity, differences in fat distribution, and gender-specific gut microbiota composition.  

 Line 452:This improvement was more pronounced in male participants than females when compared to placebo, which may suggest a hepatoprotective effect of berberine. 

The sex-specific differences may be attributed to liver enzyme activity, metabolism, and hormonal influence. Men generally have higher baseline AST levels and are more susceptible to oxidative stress, which could make them more responsive to berberine’s hepatoprotective effects. Additionally, differences in berberine metabolism and gut microbiota composition between sexes may contribute to these observations.”

Discussion Section:

1) The comparison of your study with previous studies does not include new data. Kindly use more recent studies to compare your results.

The manuscript has been updated throughout with more recent studies observing new data.

One example can be found under the discussion in lines 319 to 322.

2) Please add a paragraph discussing the potential reasons for the observed sex-specific effects, such as differences in liver enzyme activity or hormonal influences.

This has been added to the discussion as mentioned before. Please see lines 345-349:

“The differences between male and female participants may reflect sex-specific variations in lipid metabolism, possibly influenced by hormonal factors. However, the 30-day duration may have been insufficient to observe significant changes, and individual metabolic variability could also play a role.”

And Lines 399-404:

The sex-specific differences may be attributed to liver enzyme activity, metabolism, and hormonal influence. Men generally have higher baseline AST levels and are more susceptible to oxidative stress, which could make them more responsive to berberine’s hepatoprotective effects. Additionally, differences in berberine metabolism and gut microbiota composition between sexes may contribute to these observations. 

3) Additionally, there is no mention of the potential economic impact of LipoMicel Berberine, particularly in comparison to other berberine formulations.

We added a new paragraph to the discussion, please see lines 433-439:

“Beyond its potential safety and efficacy advantages, Berberine LipoMicel® enhanced bioavailability may offer economic benefits by allowing for lower dosing while achieving similar or superior therapeutic effects compared to standard berberine formulations. Traditional berberine supplements often require high doses (e.g., 1,000–1,500 mg per day) due to poor absorption, which can increase cost and dosing burden for consumers. In contrast, improved absorption with Berberine LipoMicel® may reduce the required dosage, potentially enhancing cost-effectiveness and patient compliance.”

Conclusion Section:

The conclusion should be strengthened by briefly summarizing the key findings and highlighting the implications of the study for the safety of LipoMicel Berberine®.

We revised the conclusions, please see lines 441:

“Overall, Berberine LipoMicel® was well tolerated over the 30-day study period, with no reported adverse events. All tested markers remained within recommended limits, and no statistically significant or clinically meaningful deviations were observed in blood chemistry. These findings provide strong evidence that the enhanced bioavailability of the LipoMicel® delivery system does not compromise safety in healthy individuals.

Additionally, preliminary sex-specific differences in metabolic responses were observed. Female participants had significantly lower total cholesterol (TC) levels after 30 days of Berberine LipoMicel® treatment. Although not significant, male participants exhibited a greater reduction in LDL and AST levels, while female participants showed more pronounced improvements in triglycerides, fasting blood glucose and HbA1c. These trends align with existing research suggesting that berberine’s metabolic effects may vary between genders due to hormonal and physiological differences.

These findings reinforce the safety profile of Berberine LipoMicel® and suggest potential metabolic benefits that warrant further investigation. While this study provides important initial data, further long-term, large-scale studies are needed to confirm these sex-specific responses and explore their clinical implications.”

Reviewer 2 Report

Comments and Suggestions for Authors

General comments

  • Based on the aim of this work and also on the obtained results (sometimes significant or not), can we come up with a conclusion concerning the “safety and tolerability of Berberine LipoMicel”?
  • With the small number of participants, what was the plan in case of having some bias within them (someone who doesn’t master very well your exclusion criteria).
  • Do you have references about your treatment method?
  • Are your results statistically representative even if they are statistically significant or no?

Specific comments

Line 15-16  Are 19 healthy participants enough to have significant results for this study?

Line 89   Approximately xx 1mL (check the space).

Line 91  …short 30 day xx observation period (space)

Line 139  Why did you choose the range [21-65]?

Line 174-175  What was your insurance that the participants were arriving after an 8-hour fast?

Line 182  Please make sure that your figure is clear.

Line 236  On which base did you decide to take these different numbers of participants?

Comments on the Quality of English Language

 The English could be improved to more clearly express the research.

Author Response

We appreciate the reviewer’s thoughtful feedback and constructive suggestions, which have helped strengthen our manuscript.

General comments

  • Based on the aim of this work and also on the obtained results (sometimes significant or not), can we come up with a conclusion concerning the “safety and tolerability of Berberine LipoMicel”?

Response: We have revised the conclusions of this study, please see lines 441:

“Overall, Berberine LipoMicel® was well tolerated over the 30-day study period, with no reported adverse events. All tested markers remained within recommended limits, and no statistically significant or clinically meaningful deviations were observed in blood chemistry. These findings provide strong evidence that the enhanced bioavailability of the LipoMicel® delivery system does not compromise safety in healthy individuals.

Additionally, preliminary sex-specific differences in metabolic responses were observed. Female participants had significantly lower total cholesterol (TC) levels after 30 days of Berberine LipoMicel® treatment. Although not significant, male participants exhibited a greater reduction in LDL and AST levels, while female participants showed more pronounced improvements in triglycerides, fasting blood glucose and HbA1c. These trends align with existing research suggesting that berberine’s metabolic effects may vary between genders due to hormonal and physiological differences.

These findings reinforce the safety profile of Berberine LipoMicel® and suggest potential metabolic benefits that warrant further investigation. While this study provides important initial data, further long-term, large-scale studies are needed to confirm these sex-specific responses and explore their clinical implications.”

  • With the small number of participants, what was the plan in case of having some bias within them (someone who doesn’t master very well your exclusion criteria).

Given the small sample size, we performed rigorous pre-screening to minimize bias and ensure adherence to exclusion criteria. Participants were selected based on strict inclusion/exclusion criteria— verified through medical history questionnaires and interviews; Any uncertainties regarding eligibility were resolved by study personnel before randomization.

To minimize bias from participant variability, we employed a blinded crossover design, allowing each participant to serve as their own control. This significantly reduces the impact of individual differences.

  • Do you have references about your treatment method?

Our treatment method was designed to align with Health Canada’s clinical study requirements for evaluating the safety of a new formulation. While no single reference directly outlines our exact protocol, we adhered to established regulatory guidelines and industry best practices for safety assessment, including: participant screening and monitoring to ensure adherence to eligibility criteria; blinded, placebo-controlled, crossover design to minimize bias and enhance internal validity; standardized blood chemistry panels and safety markers commonly used in clinical trials for evaluating metabolic and hepatic effects.

Given the novelty of the LipoMicel Berberine formulation, this study intends to provide important first insights into its safety profile, following regulatory standards.

  • Are your results statistically representative even if they are statistically significant or no?

We acknowledge that while our findings are statistically significant, the study sample may not be fully statistically representative of the broader population due to its relatively small size and demographic composition. However, our blinded, placebo-controlled, crossover design enhances internal validity, minimizing variability and allowing each participant to serve as their own control.

To improve generalizability, future studies with larger and more diverse populations will be essential to confirm these findings and assess potential differences across various demographic groups. Despite this limitation, our results provide valuable initial safety data on LipoMicel Berberine and support the need for further research.

Please note that we address further study limitations in the revised paper (line 406)

Specific comments

Line 15-16  Are 19 healthy participants enough to have significant results for this study?

We acknowledge that the sample size of 19 healthy participants is relatively small; however, the study was designed as a placebo-controlled, blinded, crossover trial, which strengthens statistical power by allowing each participant to serve as their own control. Please find more information about sample size calculations in the paper, as outlined in Section 2.6 (Sample Size) of the manuscript.

However, we recognize the importance of larger, long-term studies to further confirm these findings and improve generalizability.

Line 89   Approximately xx 1mL (check the space).

This is now line 96 and the space has been erased.

Line 91  …short 30 day xx observation period (space)

This is now line 98 and the space has been erased.

Line 139  Why did you choose the range [21-65]?

(Now line 150)

To our knowledge, the 21-65 age range is commonly used in safety studies as it minimizes variability by excluding adolescent <21 and older adults >65 who often present with comorbidities or age-related metabolic changes, and polypharmacy. Furthermore, this range aligns with regulatory guidelines.

Line 174-175  What was your insurance that the participants were arriving after an 8-hour fast?

(Now lines 199-200)

To ensure compliance with the 8-hour fasting requirement, we implemented the following measures:

  1. Pre-Visit Instructions: Participants were provided with detailed verbal and written instructions emphasizing the importance of an 8-hour fast prior to each visit. These instructions specified that only water was permitted during the fasting period.
  2. Confirmation Prior to Sample Collection: Upon arrival, participants were asked to verbally confirm their adherence to the fasting protocol. Those who reported any deviation from the fasting requirement were rescheduled.
  3. Incentive for Compliance: Participants were incentivized to observe an overnight fast by the provision of a standard breakfast following their blood draw, further encouraging adherence to the fasting protocol.
  4. Biochemical Markers as Indicators of Compliance: Parameters such as blood glucose (measured but not reported), triglycerides, HDL, LDL, and total cholesterol were analyzed, as these markers would show significant elevation if participants had not observed the fasting period. No unexpected elevations in these parameters were observed in our data, further supporting that participants complied with the fasting requirement.
  5. Morning Appointments: To minimize the likelihood of non-compliance, all sample collections were scheduled in the morning, and participants were instructed to abstain from consuming food or beverages (other than water) after 10 PM the night before.

These measures were implemented to ensure participant compliance and maintain the validity of the study data.

Line 182  Please make sure that your figure is clear.

Figure 1 has now been modified for clarity.

Line 236  On which base did you decide to take these different numbers of participants?

(Now line 257)

Our study screened a total of 19 participants for eligibility. After the initial screening, 19 participants were found to meet the inclusion and exclusion criteria and were willing to give their informed consent. Of the 19 participants, 12 were male and 7 were female. Our sample size calculations (Section 2.7.) indicated that these numbers would be sufficient for the evaluation of various blood chemistry parameters based on the study design (placebo-controlled, crossover).

Comments on the Quality of English Language

 The English could be improved to more clearly express the research.

Thank you for your feedback. The manuscript has been thoroughly revised for clarity and readability, including a review by native English speakers. We have carefully refined the text to ensure clearer expression of the research findings.

Reviewer 3 Report

Comments and Suggestions for Authors

it is an intriguing article talking about the safety and tolerability of new formulation barberine with better bioavailability. 

1) Introduction section, line 57 -68, any references? 

2) any explanation regarding to decrease of Hb1Ac in female participants? it might be helpful to discuss in Discussion section.

3) it might be interesting to discuss that any other formulation barberine's capability of inductions in cholesterol and H1bA1c

Author Response

We appreciate the reviewer’s thoughtful feedback and constructive suggestions, which have helped strengthen our manuscript.

Comments: It is an intriguing article talking about the safety and tolerability of new formulation berberine with better bioavailability. 

1) Introduction section, line 57 -68, any references? 

Response: This has been revised to include references in the manuscript (now lines 64-75)

2) any explanation regarding the decrease of Hb1Ac in female participants? it might be helpful to discuss in Discussion section.

Response: This has been revised and addressed in the manuscript, please see the following:

Line 324: “Interestingly, female participants in the placebo group had significantly lower fasting glucose and HbA1c concentrations compared to those receiving berberine treatment.  A possible explanation for this could be a carryover effect, suggesting that the washout period may not have been sufficient to fully eliminate the effects of berberine before participants crossed over to the placebo phase. Given berberine's potential to influence glucose metabolism over time, residual effects from the treatment phase could have contributed to the observed glucose and HbA1c differences.”

Line 363:  “Regarding blood sugar modulation, it should be noted that HbA1c typically measures the 3-month averages of blood sugar levels whereas the changes observed in the participants were over a week-to-week basis spanning only a month. Therefore, HbA1c cannot reliably reflect glycemic improvement in 30 days. Nevertheless, female participants showed a more noticeable downward trend in HbA1c compared to placebo after 30 days. In contrast, male participants did not exhibit the same trend in blood sugar reductions. This aligns with a recent meta-analysis, which concluded that berberine’s effects on glycemic control and insulin sensitivity may be more pronounced in women compared to men [33]. This may be attributed to estrogen’s role in enhancing insulin sensitivity, differences in fat distribution, and gender-specific gut microbiota composition. 

Other potential explanations include the fact that dietary habits or lifestyle changes were not controlled for in this study. Additionally, it is important to note that all participants were healthy, normal-weight individuals, and therefore, reductions in HbA1c (as well as blood lipids) may not have been as pronounced as in previous studies on (standard) berberine, which primarily focused on overweight populations over a longer study period. Furthermore, while in a previous study comparing different berberine treatments (i.e., standard vs LipoMicel® formulation), immediate effects on fasting blood glucose (~12% reduction) were observed within just 2 days of LipoMicel® treatment in healthy participants, this was likely due to the treatment’s enhanced bioavailability ability leading to improved insulin sensitivity and glucose uptake. [23]. Similarly, in the present study, a substantial 15% decrease in fasting blood glucose (from 6.0 mmol/L to 5.1 mmol/L) was observed in female participants within the first week of treatment, indicating a meaningful early metabolic improvement. This reduction was sustained over 30 days and accompanied by an early trend toward improvement in HbA1c. However, the changes did not reach statistical significance, likely due to the small sample size. Notably, the magnitude and timing of the observed fasting glucose reduction are comparable to the early-phase glycemic response typically seen with metformin initiation [34,35]. In contrast, standard berberine formulations generally require a longer duration and higher doses to achieve similar effects, highlighting the potential advantage of the LipoMicel® formulation in delivering faster glycemic benefits.  “

3) it might be interesting to discuss that any other formulation berberine's capability of inductions in cholesterol and H1bA1c

  1. A) Regarding berberine’s effect on blood lipids like cholesterol, we addressed this in the manuscript (please see lines 352-359): “Multiple studies have already demonstrated the significant cholesterol-lowering effects of berberine (both unformulated berberine extract and berberine chloride)—highlighting it as a promising natural agent for managing dyslipidemia [28,29]. Among the suggested pharmacological mechanisms of action are the up-regulation of LDL receptors through mRNA stabilization, which is distinct from statins [30], as well as the inhibition of cholesterol absorption in the intestine i.e. by interfering with micellization, decreasing uptake and permeability, and inhibiting acyl-coenzyme A cholesterol acyltransferase-2 expression [31].
  2. B) Regarding Hb1AC, we addressed this in the following line 375-391:

“…reductions in Hb1Ac (as well as blood lipids) may not have been as pronounced as in previous studies on (standard) berberine, which primarily focused on overweight populations over a longer study period. Furthermore, while in a previous study comparing different berberine treatments (i.e., standard vs LipoMicel® formulation), immediate effects on fasting blood glucose (~12% reduction) were observed within just 2 days of LipoMicel® treatment in healthy participants, this was likely due to the treatment’s enhanced bioavailability ability leading to improved insulin sensitivity and glucose uptake. [23]. Similarly, in the present study, a substantial 15% decrease in fasting blood glucose (from 6.0 mmol/L to 5.1 mmol/L) was observed in female participants within the first week of treatment, indicating a meaningful early metabolic improvement. This reduction was sustained over 30 days and accompanied by an early trend toward improvement in HbA1c. However, the changes did not reach statistical significance, likely due to the small sample size. Notably, the magnitude and timing of the observed fasting glucose reduction are comparable to the early-phase glycemic response typically seen with metformin initiation [34,35]. In contrast, standard berberine formulations generally require a longer duration and higher doses to achieve similar effects, highlighting the potential advantage of the LipoMicel® formulation in delivering faster glycemic benefits. “ 

Round 2

Reviewer 1 Report

Comments and Suggestions for Authors

In light of my comments at the Introduction section, it would be better for the authors to emphasize their study and confirm berberine's status as a medicinal plant, similar to other plants discussed.

Author Response

We thank the reviewer for their time and constructive comment, which have helped to widen the appeal of our manuscript.

Comments:

In light of my comments at the Introduction section, it would be better for the authors to emphasize their study and confirm berberine's status as a medicinal plant, similar to other plants discussed.

Response: Thank you! Accordingly, we have updated the introduction as follows:

Brief introduction regarding use of other medicinal plants added at Lines 41-44:

Plants have a long history of use in the form of powders, teas, poultices, infusions, and essential oils for their medicinal properties in various traditional cultures, and many modern day natural health products consist of medicinal ingredients (MI) purified from traditional plants [1,2].

Contribution of the current study to the usage of berberine as a plant-derived medicinal ingredient at Lines 112-114:

“the purpose of this research was to confirm that Berberine LipoMicel® is safe and well-tolerated for short-term use in humans and to further support the usage of berberine as a medicinal plant ingredient.”
